

# How to normalize metatranscriptomic count data for differential expression analysis

Heiner Klingenberg and Peter Meinicke

Department of Bioinformatics, Institute of Microbiology and Genetics, University of Goettingen, Göttingen, Germany

## ABSTRACT

**Background**. Differential expression analysis on the basis of RNA-Seq count data has become a standard tool in transcriptomics. Several studies have shown that prior normalization of the data is crucial for a reliable detection of transcriptional differences. Until now it has not been clear whether and how the transcriptomic approach can be used for differential expression analysis in metatranscriptomics.

**Methods**. We propose a model for differential expression in metatranscriptomics that explicitly accounts for variations in the taxonomic composition of transcripts across different samples. As a main consequence the correct normalization of metatranscriptomic count data under this model requires the taxonomic separation of the data into organism-specific bins. Then the taxon-specific scaling of organism profiles yields a valid normalization and allows us to recombine the scaled profiles into a metatranscriptomic count matrix. This matrix can then be analyzed with statistical tools for transcriptomic count data. For taxon-specific scaling and recombination of scaled counts we provide a simple R script.

**Results**. When applying transcriptomic tools for differential expression analysis directly to metatranscriptomic data with an organism-independent (global) scaling of counts the resulting differences may be difficult to interpret. The differences may correspond to changing functional profiles of the contributing organisms but may also result from a variation of taxonomic abundances. Taxon-specific scaling eliminates this variation and therefore the resulting differences actually reflect a different behavior of organisms under changing conditions. In simulation studies we show that the divergence between results from global and taxon-specific scaling can be drastic. In particular, the variation of organism abundances can imply a considerable increase of significant differences with global scaling. Also, on real metatranscriptomic data, the predictions from taxon-specific and global scaling can differ widely. Our studies indicate that in real data applications performed with global scaling it might be impossible to distinguish between differential expression in terms of transcriptomic changes and differential composition in terms of changing taxonomic proportions.

**Conclusions**. As in transcriptomics, a proper normalization of count data is also essential for differential expression analysis in metatranscriptomics. Our model implies a taxon-specific scaling of counts for normalization of the data. The application of taxon-specific scaling consequently removes taxonomic composition variations from functional profiles and therefore provides a clear interpretation of the observed functional differences.

Corresponding author
Peter Meinicke, peter@gobics.de

# BACKGROUND

Metagenome analysis can provide a comprehensive view on the metabolic potential of a microbial community (*Eisen, 2007*; *Simon & Daniel, 2009*). In addition to the static functional profile of the metagenome, metatranscriptomic RNA sequencing (RNA-Seq) can highlight the multi-organism dynamics in terms of the corresponding expression profiles (*Poretsky et al., 2005*; *Frias-Lopez et al., 2008*; *Gilbert et al., 2008*; *Urich et al., 2008*). In particular, metatranscriptomics makes it possible to investigate the functional response of the community to environmental changes (*Gilbert et al., 2008*; *Poretsky et al., 2009*).

In single organism transcriptome studies, differential expression analysis based on RNA-Seq data has become an established tool (*Marioni et al., 2008*; *Trapnell et al., 2012*). For the analysis, first, quality-checked sequence reads are mapped to the organisms genome for transcript identification. Then the transcript counts are compared between different experimental conditions to identify statistically significant differences. Several studies have shown that read count normalization has a great impact on the detection of significant differences (*Bullard et al., 2010*; *Dillies et al., 2013*; *Lin et al., 2016*). The aim of the count normalization is to make the expression levels comparable across different samples and conditions. This is an essential prerequisite for distinguishing condition-dependent differences from spurious variation of expression levels.

In metatranscriptomics, already the transcript identification step can be challenging. In many cases, RNA-Seq faces a mixture of organisms for which no reference genome sequence is available. Several strategies have been suggested: de novo transcriptome assembly combined with successive homology-based annotation (*Celaj et al., 2014*), the direct functional annotation of reads by classification according to some protein database (*Huson et al., 2011*; *Nacke et al., 2014*; *Hesse et al., 2015*) or parallel sequencing of the corresponding metagenome with successive mapping of RNA-Seq reads to assembled and annotated contigs (*Mason et al., 2012*; *Franzosa et al., 2014*; *Ye & Tang, 2016*). For the subsequent comparison of counts between different conditions no standard protocol exists for differential expression analysis on metatranscriptomic data. Several studies and tools apply methods that have been developed for differential expression analysis in transcriptomics to metatranscriptomic count data (*McNulty et al., 2013*; *Martinez et al., 2016*; *Macklaim et al., 2013*). However, the question of under which conditions established models from single organism transcriptomics also apply to organism communities has not been addressed sufficiently so far.

Here we present an extended statistical model for count data from metatranscriptomic RNA-Seq experiments. The model provides a novel view on metatranscriptome data, which negates the impact of taxonomic abundance differences between samples for differential expression analysis. Theoretical considerations, as well as studies on simulated and real count data, show that our approach can help to identify differentially expressed features

which actually reflect a changing behavior of the organisms. For our model to work, accurate normalization of the data is crucial and in general, requires an organism-specific rescaling of expression profiles. The application of differential expression analysis to mixed-species data without prior separation can be found in several metatranscriptomic studies (*Nacke et al., 2014*; *McNulty et al., 2013*; *De Filippis et al., 2016*) as well as in dedicated pipelines for metatranscriptome analysis (*Martinez et al., 2016*; *Westreich et al., 2016*). Our results suggest that this approach may imply a large number of significant differences which lack a clear interpretation in terms of functional responses.

## MATERIALS & METHODS

### Normalization in transcriptomics

To clarify our arguments for an alternative normalization of metatranscriptomic data we need to explain the statistical nature of the normalization problem. We first follow the approach of *Anders & Huber (2010)* for single organism RNA-Seq count data and start with a basic model for the mean of the observed counts. The expected (mean) count $\mathbb{E}[Y_{ij}]$ for gene (feature) $i$ and sample $j$ arises from a product of the per-gene quantity $\lambda_{ic_j}$ under condition $c_j$ and a size factor $s_j$:

$$\mathbb{E}[Y_{ij}] = \lambda_{ic_j} s_j. \tag{1}$$

The factor $\lambda_{ic_j}$ is proportional to the mean concentration of feature $i$ under condition $c_j$. The size factor $s_j$ represents the sampling depth or library size. Usually, both factors are unknown. If we assume the $i$th feature to be non-differentially expressed (NDE) we can represent the corresponding row of the count matrix by

$$\mathbb{E}\left[Y_{i\bullet}^{(\mathrm{NDE})}\right] = \lambda_i \mathbf{s} \tag{2}$$

where the relative feature abundance is equal for all samples and all size factors have been comprised in the row vector $\mathbf{s}$. Thus, for NDE features the size factors are proportional to the expected counts. If we knew which features are actually NDE, we would be able to estimate the required size (scaling) factors for normalization from the corresponding counts.

Usually this is not the case and we need to make some assumptions. A common assumption that is used in current tools is that most of the features are NDE. Then it is possible to estimate the scaling factors by some robust statistics. In DESeq for each sample the putative scaling factors from all features are calculated and then the median of all these values is used as an estimator of the sample-specific scaling factor (*Anders & Huber, 2010*). With a breakdown point of 50% the median requires that the majority of the data corresponds to NDE features.

Without any distinction between DE and NDE features the scaling factors have also been estimated from the count sums of all samples. However, the potential shortcomings of this total count normalization have widely been discussed (*Anders & Huber, 2010*; *Robinson & Oshlack, 2010*; *Soneson & Delorenzi, 2013*).

## Normalization in metatranscriptomics

In metatranscriptomics the situation is more complicated because for each organism we can have a different scaling factor. So we have to extend the above sampling model to an $N$-organism mixture that includes a matrix $\mathbf{S}$ of organism-specific scaling factors $s_{jk}$:

$$\mathbb{E}\left[Y_{ij}\right] = \sum_{k=1}^{N} \lambda_{ijk} s_{jk} \tag{3}$$

where $i, j, k$ are the feature, sample and organism indices, respectively. We omitted the condition dependency ($c_j$) for convenience.

In analogy to Eq. (2) for NDE features we have the following model for a feature row $i$ of the count matrix:

$$\mathbb{E}\left[Y_{i\bullet}^{(\text{NDE})}\right] = \boldsymbol{\lambda}_i^T \mathbf{S}^T \tag{4}$$

where the column vector $\boldsymbol{\lambda}_i$ contains all organism-specific rates for feature $i$ and $\boldsymbol{\lambda}_i^T$ indicates transposition of this vector.

Application of the above single-organism scheme for estimation of scaling factors is only valid if the matrix of scaling factors has the following form:

$$\mathbf{S} = [\alpha_1 \mathbf{s}, \alpha_2 \mathbf{s}, \ldots, \alpha_K \mathbf{s}] = \mathbf{s}\boldsymbol{\alpha}^T \tag{5}$$

where $\boldsymbol{\alpha}$ is a column vector of organism-specific abundances and $\mathbf{s}$ contains the sample-specific scaling factors, now in a column vector, which is equal for all organisms. Then we can write

$$\mathbf{S}\boldsymbol{\lambda}_i = \mathbf{s}\boldsymbol{\alpha}^T \boldsymbol{\lambda}_i = \tilde{\lambda}_i \mathbf{s} \tag{6}$$

where $\tilde{\lambda}_i$ results from the dot product of the organism and the feature rates. This corresponds to Eq. (2) and allows to apply DESeq or other tools for single organism differential expression analysis to the metatranscriptomic count matrix. However, the underlying assumption that $\mathbf{S}$ has column rank 1, i.e., all column vectors are collinear, would be hard to justify in practice. Implicitly we would assume that the relative contributions of all organisms are constant across all samples. In general, this assumption is not met, because for a real metatranscriptome, the organism composition of transcripts cannot be controlled and will be different for different samples.

In the following we show how to normalize metatranscriptomic counts so that the data actually meet the former assumption.

## Taxon-specific scaling and global scaling

We propose a method to prepare metatranscriptomic data for differential expression analysis. The method is referred to as taxon-specific scaling. As an essential prerequisite our approach requires that the data is first partitioned according to the contributing organisms. Then the count data matrix from each partition is normalized separately. Here, established tools from transcriptomics can be used to estimate the corresponding scaling factors.

Finally, the normalized count data matrices are summed up to provide normalized metatranscriptomic count data which can be analyzed in terms of differential expression

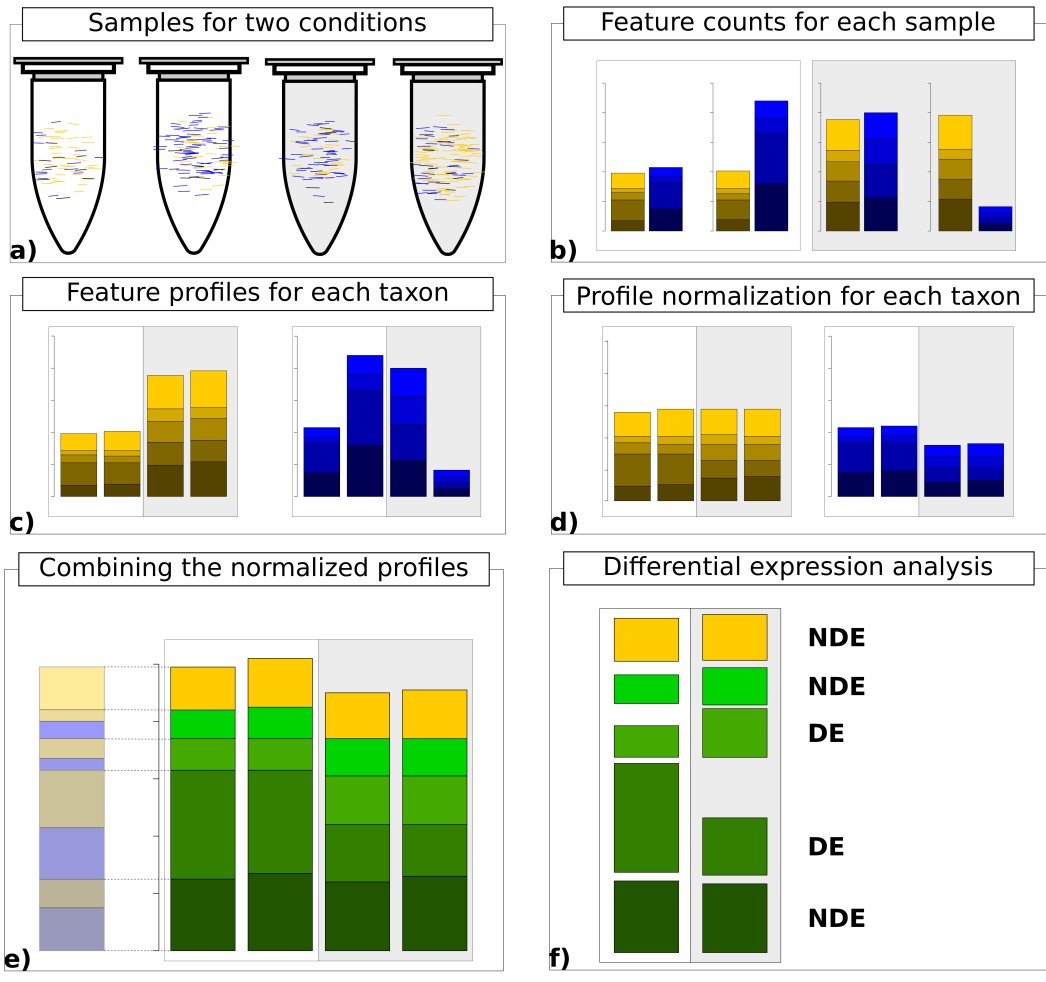

**Figure 1   Work flow for taxon-specific normalization.** (A) Sequence samples from conditions A (white) and B (light gray). Assign each sequence read to taxonomic and feature categories. (B) Compute feature profiles from the assignment counts. (C) Obtain count matrix from taxon-specific feature profiles. (D) Normalize feature profiles of each taxon-specific count matrix separately. (E) Recombine normalized feature profiles of all taxa into a metatranscriptomic profile. (F) Perform differential expression analysis on metatranscriptomic count matrix.

(Fig. 1). Here all statistical models and tools for count-based differential expression analysis in transcriptomics can in principle be used to identify differentially expressed features.

If we denote the original count matrix for organism $k$ as $\mathbf{Y}_k$ and the associated vector of estimated scaling factors as $\hat{\mathbf{s}}_k$ the normalized metatranscriptomic count matrix is computed by

$$\tilde{\mathbf{Y}} = \sum_k \mathbf{Y}_k \, \mathrm{diag}^{-1}(\hat{\mathbf{s}}_k). \tag{7}$$

Here, the $\mathrm{diag}^{-1}$ operator transforms the scaling vector to a diagonal matrix with inverse scaling factors on the diagonal and zeros everywhere else. We provide an R script where we

use DESeq2 for scaling factor estimation and identification of significant differences (see File S1).

In principle, our method is computationally simple and the hard work has to be done beforehand in order to provide the partitioned data in terms of the organism-specific count matrices. This is the realm of binning methods and, in addition, may require sequence assembly tools to achieve a sufficient sequence length for reliable separation.

At this point, the question may arise as to why get back to metatranscriptomic data when differential expression analysis could be performed for separate organisms or specific taxa. In this context, by metatranscriptomic counts we refer to mixed counts that arise from a sum over superimposed organisms. There are several reasons why the analysis of the recombined metatranscriptome data can be useful: first of all, the statistical power of organism-specific tests may be low due to decreased counts. If several organisms show the same slight difference, this difference may only become statistically significant when accumulating their normalized counts. Or a feature may show differences for single organisms but these differences may cancel out when correctly summarized. In this case the corresponding feature is not indicative for the experimental condition with regard to the whole community. Therefore the analysis of separate organism transcriptomes and the analysis of the rectified metatranscriptome data should be combined to provide a complete picture of the community response.

In our study and in the supplied R script we use DESeq2 to compute scaling factors and to identify significant differences on the basis of the normalized count matrices. We decided for DESeq2 for several reasons. It is an established tool in transcriptomics which has shown a good performance in comparative studies (*Soneson & Delorenzi, 2013*; *Dillies et al., 2013*) and which has already been used for metatranscriptome analysis (*McNulty et al., 2013*; *Martinez et al., 2016*; *De Filippis et al., 2016*). In particular, the estimation of scaling factors is robust and can be performed as a separate prior step apart from the computation of significant differences. The latter aspect is important for taxon-specific scaling which requires applying the normalization independently. However, we would like to emphasize that our arguments for the taxon-specific scaling approach do not depend on a particular statistical tool and in fact the main findings of our study can be reproduced with other tools, such as edgeR, SAMseq (*Li & Tibshirani, 2013*) or limma (*Ritchie et al., 2015*). In some experiments we also used edgeR and total count (TC) normalization to study the impact of different transcriptomic scaling methods.

In contrast to taxon-specific scaling, global scaling performs the normalization of metatranscriptomic data without prior separation, i.e., sample-specific scaling factors are estimated from the original metatranscriptomic counts. In general, taxon-specific and global scaling will result in distinct normalized count matrices which in turn can lead to largely differing results in differential expression analysis.

Taxon-specific scaling and global scaling provide two different views on the metatranscriptome. Taxon-specific scaling considers varying taxonomic abundances as a confounding factor and eliminates its influence on the identification of differentially expressed features (DEF) by scaling the profiles of each organism individually. In contrast, global scaling normalizes the data with respect to the total library size, i.e.,

the sample-specific count sum, and taxonomic abundance shifts can strongly influence the identification of DEF. With global scaling, the researcher expects the taxonomic composition of the samples to be informative for differential expression analysis. However, it is not always clear how this information can be utilized. When relative taxonomic abundance varies within a condition, this implies a higher dispersion of the corresponding count data, which in turn may prevent the identification of DEF. Furthermore, if DEF are identified, it is unclear whether they result from organisms which show a differing functional profile between conditions or whether they result from organisms with unchanged functional profiles but with varying relative abundances. Finally, if taxonomic composition variations in the data are maintained it may happen that for organisms that are only abundant in one condition all of their associated functions become differentially expressed. If this affects the majority of features, the model assumptions of statistical tools for DEF identification may be violated (see above).

Normalization methods have also been used in the field of metagenomics to make the taxonomic compositions between samples comparable. Here, similar techniques can be applied (*McMurdie & Holmes, 2014*; *Weiss et al., 2017*) to identify the differential abundance of taxons. However, while in metagenomics the number of reads for a taxon is proportional to the genome size (or marker gene copy number) and the abundance of the organism, in metatranscriptomics, each gene can have a different transcription rate. When using tools like edgeR (*Robinson, McCarthy & Smyth, 2010*) or DESeq2 (*Love, Huber & Anders, 2014*) for the identification of differential taxonomic abundance in metagenomic data, the model assumptions may be violated. In particular, it is unclear if the majority of taxons do not change between different communities.

## Simulated metatranscriptome

The aim of our simulations was to demonstrate and explain the differences between taxon-specific and global scaling in practice, using a fully controlled data generation setup. To make the differences visible and understandable it is sufficient to simulate small communities as an increased complexity requires more parameters to be controlled which may even complicate interpretation. In some cases even a minimal community with two organisms may be sufficient to clearly show the different effects of the two scaling methods.

In our simulations, a metatranscriptome arises from a mixture of various organisms, each with individual features. We assume that all organisms somehow react to a change of experimental conditions in terms of changing feature profiles. A metatranscriptome can include features covered by all taxa as well as features occurring only in few or a single organism. The sum of contributed counts from a single organism we refer to as the organism-specific library size. Generally, the count contributions from different organisms are not equal and vary across samples. We refer to this as the variation of the library size. To simulate the variation of organism-specific library sizes, we generated multiple transcriptomic count data sets with different total count numbers. Thereby, each generated data set mimics the contribution of a single organism. The data sets were then combined to simulate a metatranscriptomic count matrix.

As with all simulations, the generated data can only provide a coarse approximation of real data and the obtained counts depend on particular parameters. Therefore, settings for

the number of features and the total count values influence the results. Each organism is simulated with 100 differentially expressed features (DEF), 50 of them upregulated, and with 900 features that were non-differentially expressed (NDE).

Each data set consists of two conditions, A and B, with six samples (replicates) per condition and up to five organisms (Org1 to Org5) per sample. In the first three simulations, the different organism profiles are stacked, to exclude any interference between features from different organisms in the combined data. Accordingly, the final count matrix has 12 columns and 5,000 rows that correspond to samples and features, respectively. In our simulated metatranscriptome experiments we show under which circumstances, the differences between global scaling and taxon-specific scaling increase in respect to the number of identified DEF. With the stacked profiles we focus on the ability to recover DEF which the data generation process labeled as DE for the simulated transcriptomes. Within this context, the data generation provides the necessary information to calculate the number of true positives (TP) and false positives (FP). The label $L_i$ is DE or NDE according to feature $i$ being differentially expressed or non-differentially expressed. The statistical test used to detect DEF, provides a $p$-value for each feature. The predicted label $\hat{L}_i$ is DE if the adjusted $p$-value (*Benjamini & Hochberg, 1995*) is below a threshold of 0.05 for feature $i$. The TP and FP counts are calculated for each organism $k$ individually:

$$\text{TP}_k = |\{i : \hat{L}_i = \text{DE} \wedge L_i = \text{DE}\}| \tag{8}$$

$$\text{FP}_k = |\{i : \hat{L}_i = \text{DE} \wedge L_i = \text{NDE}\}|. \tag{9}$$

### Synthetic data generation and analysis tools

The tool compcodeR (*Soneson, 2014*) was used to generate all simulated data. The tool generates count data based on a negative binomial distribution model with parameters estimated from real transcriptome data (*Pickrell et al., 2010*; *Cheung et al., 2010*). If not explicitly specified, the compcodeR parameters in the R function "generate.org.mat" are used (see File S1). We also included the real data in csv format (File S2) and a R function "run.experiments.tax.glo" (File S3) to perform the simulations and real data analyses. All analyses were performed with R version 3.3.0, DESeq2 version 1.8.2 and edgeR version 3.10.5, Bioconductor version 3.1 and compcodeR version 1.4.0.

### Simulation I: "Without library size variation"

In the first experiment we simulate the case where the library size (LS) for each of the five organisms does not vary across different samples. Although this is an unrealistic case we performed this simulation to verify that both normalization approaches produce the same results under idealized conditions.

In addition, we wanted to investigate how different organism abundances affect the identification of DEF. Each organism was assigned a fixed total count number across all samples, without variation in library size. We simulated organism Org1 with a base count of $1e^7$ followed by organism Org2 with $5e^6$, $1e^6$ for organism Org3 and organism Org4, Org5 with $5e^5$, $1e^5$ respectively. This results in relative abundance fractions of 0.60, 0.30, 0.06, 0.03 and 0.01 for the 5 organisms. Because data is generated without variation for the number of counts per sample, no normalization is required, i.e., the correct scaling factors for all samples and all organisms are the same (=1).

### Simulation II: "Variations in library size of species present in the metacommunity"

In the second experiment we simulated a more realistic situation, with varying LS for all included organisms. Organism base counts are identical to the first simulation but the LS is randomly increased or reduced according to a random factor between 0.5 and 2. Due to the different library sizes in the samples, a prior normalization for both global scaling and taxon-specific scaling is required.

### Simulation III: "Condition dependent variation"

In the third simulation, we investigated to what extent a condition dependent variation of LS can affect the normalization results. Under condition A we increase LS of Org1 by a random factor between 1.5 and 2 while under condition B we decrease the LS by a random factor between 0.5 and 0.667. For Org2 the direction of change is reversed, with a random decrease under condition A and an increase for condition B. For Org1 and Org2 the same base count as for Simulation I and II is used and for Org3–5 all parameters from Simulation II are used.

### Simulation IV: "Mixed feature effects"

In this simulation, we investigated the different effects that can be observed for taxon-specific scaling and global scaling when using superimposed (mixed) counts. For the superposition, we assume that the corresponding counts of equal functional categories are summed up across different species.

For an easier visualization, we reduced the setup to two organisms with a base count of 1e7 for both. Here, we used the LS variation range for Org1 and Org2 from Simulation III. CompcodeR generated each organism profile with first 50 features as upregulated DEF, next 50 as downregulated DEF, followed by additional 900 NDE features. As a result, we obtain a total of 1,000 features for the superimposed data. For the mixed features, we sum up the corresponding features ($a$ for Org1 and $b$ for Org2) of both organisms i.e., feature $a_1 + b_1, a_2 + b_2, \ldots, a_{1,000} + b_{1,000}$. Thus, the feature combination results in summing up DEF with the same direction for both organisms.

### Scaling factor divergence

The scaling divergence $D_k$ was estimated from the difference between the sample-specific scaling factors $\hat{s}_{jk}$ and the actual scaling factors $s_{jk}$ for each organism $k$ as provided by the simulation parameters. In both cases the factors are scaled to provide a unit mean across samples. To obtain a value between 0 and 1, we compute the divergence by:

$$D_k = \frac{\sum_j |\hat{s}_{jk} - s_{jk}|}{2n} \tag{10}$$

where $n$ is the number of samples. In addition, we used the logarithmic measure $\log_2 \hat{s}_{jk}/s_{jk}$ to represent the directed divergence.

## Metatranscriptome data

For a real data study, we chose a metatranscriptome dataset from mice gut (*McNulty et al., 2013*). The experiment includes 12 different species (see File S4: Table 1) representing an

artificial human gut microbial community which was inserted into germ free mice. In the original study the diet for the mice was changed at different time points. Metatranscriptomic data is available for six time points which provide the conditions for our analysis. The available processed count data was obtained from the European Bioinformatics Institute (http://www.ebi.ac.uk/, ArrayExpress, E-GEOD-48993) and contains gene names and the associated numbers.

Because the gene to Pfam (*Finn et al., 2014*) mapping is available for most organisms, we selected Pfam protein domains as features for the differential expression analysis. Each Pfam domain family is a feature in the resulting vector, including only Pfams observed at least once. We transformed the available RPKM values for the genes back to raw counts. For genes with multiple Pfam annotations, we added the raw count values of the gene to all associated Pfam features. From the available data, we constructed a count matrix for each condition and organism (File S5). Here, each column constitutes a different sample and each row represents a particular feature. Because the count data from *Bacteroides cellulosilyticus WH2* did not map to gene names, all related counts are excluded from the analysis.

A differential expression analysis for all pairwise combinations of distinguished conditions was performed to compare the results of global and taxon-specific scaling. We calculated the number of DEF predicted (a) with both methods with the same fold change direction, (b) with both methods but with an opposite fold change direction and (c) with only one scaling method. In addition, we investigated the overlap between the single organism transcriptome analyses and the differential expression analysis for the mixture. We applied a significance threshold on the adjusted *p*-value of 0.05 for the prediction of DEF.

## RESULTS

In the first part of our evaluation we examined the performance of taxon-specific and global scaling methods on simulated data. Because simulations I to III had been designed to provide a clear ground truth from a transcriptomics perspective we were able to distinguish true positive predictions of DEF from falsely classified features. In the second part we show results on real metatranscriptomic count data. Here the ground truth is not known and therefore we restrict the analysis on the comparison of the results from the two normalization approaches. Because it is impossible to verify the correctness of predictions we focus on analyzing the agreement or disagreement on DEF detection in this case.

### Simulation I

In this experiment, we measured the ability to detect DEF in a metatranscriptome without variation of organism-specific library sizes across different samples. This situation, in principle, does not require any normalization and therefore we expected taxon-specific ("tax") and global ("glo") scaling to yield similar results. This is confirmed by the resulting true positive (TP) predictions of DEF for the included organisms (Fig. 2). For both approaches the number of true positives is higher for more abundant organisms due to an increased statistical power of the corresponding tests. The final profile includes 100 DE and 900 NDE features for each organism, resulting in 5,000 features in total.

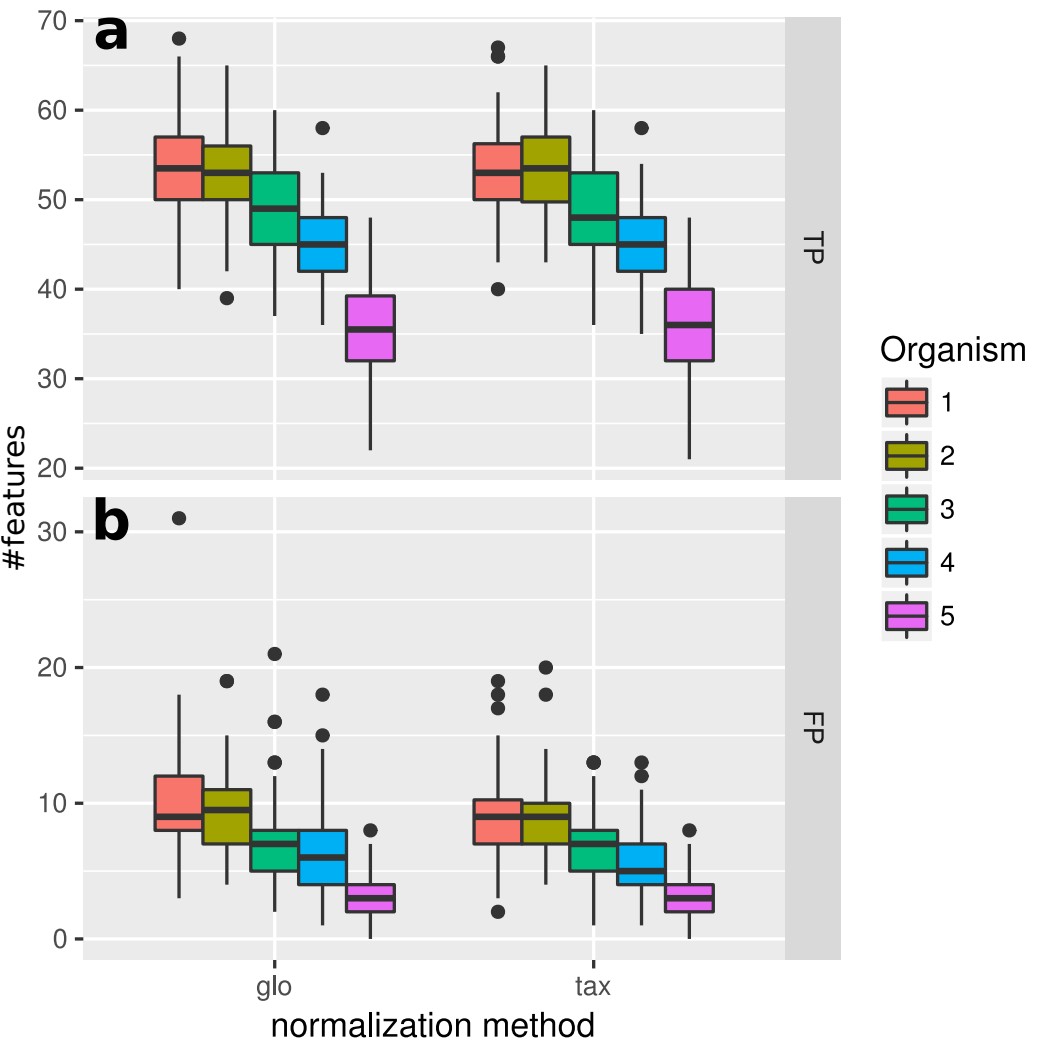

**Figure 2** **Simulation I.** Number of true positive (TP) and false positive (FP) features identified with DE-Seq2 for global ("glo") and taxon-specific ("tax") scaling: boxplots represent variation over 100 runs of the simulation.

We repeated the analysis with edgeR with the included normalization and DESeq2/edgeR with TC normalization to estimate the scaling factors (see File S4: Fig. 1). For edgeR, the number of correctly identified DEF is lower for all organisms (see File S4: Fig. 1). For this particular data set, the library size (LS) was correctly adjusted by DESeq2 with scaling factors close to 1 for all samples. Again both normalization approaches performed equally well. A similar picture can be expected for a varying total LS of the metatranscriptome samples as long as the relative LS of the organisms does not vary across different samples.

### Simulation II

When introducing organism-specific LS variation across samples the picture changes. For the global scaling approach the results show a decrease in the average TP rate for all organisms (see Fig. 3). This trend is also visible when edgeR with the included normalization

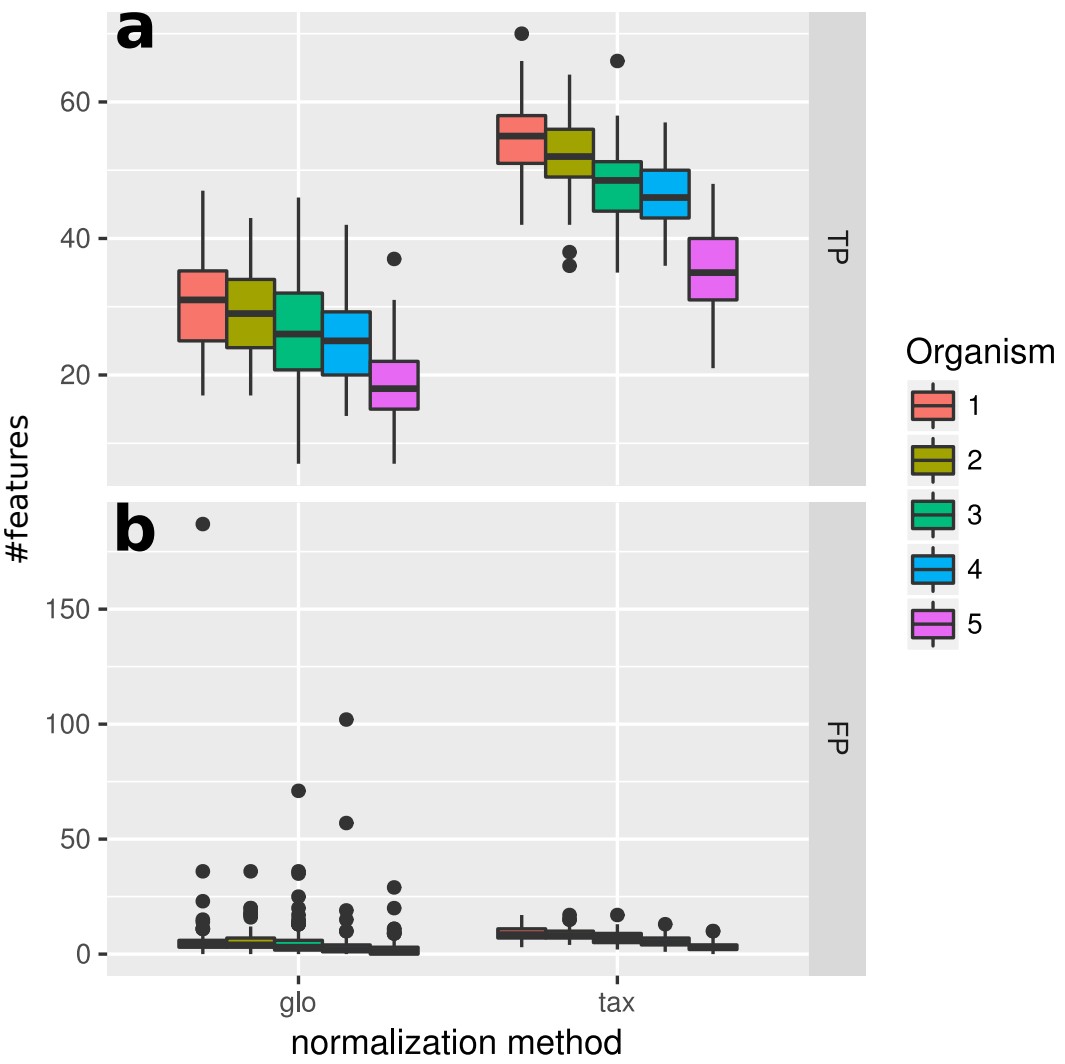

**Figure 3 Simulation II.** Number of true positive (TP) and false positive (FP) features identified with DE-Seq2 for global ("glo") and taxon-specific ("tax") scaling. FP boxplots appear compressed due to outliers. Organism order for FP is the same as for TP boxplots.

and DESeq2/edgeR with TC normalization are used for differential expression analysis (see File S4: Fig. 2). On the other hand, with taxon-specific scaling the results are very similar to results from Simulation I (see Fig. 2). With this method more DEF are correctly identified than with global scaling. The difference in the number of correctly identified DEF for global scaling is dependent on the parameter settings for the LS variation. For a lower amplitude of the LS variation, the TP rate for global scaling increases (File S4: Fig. 3). The range of TP for the most abundant organisms Org1 and Org2 is broader with global scaling (see Fig. 3) which also shows a higher scaling divergence for organisms with a lower sequencing depth (see File S4: Fig. 4).

The receiver operating characteristics (ROC) shows a higher area under curve (AUC) value for taxon-specific scaling (0.8776) than for global scaling (0.8282). The curve for

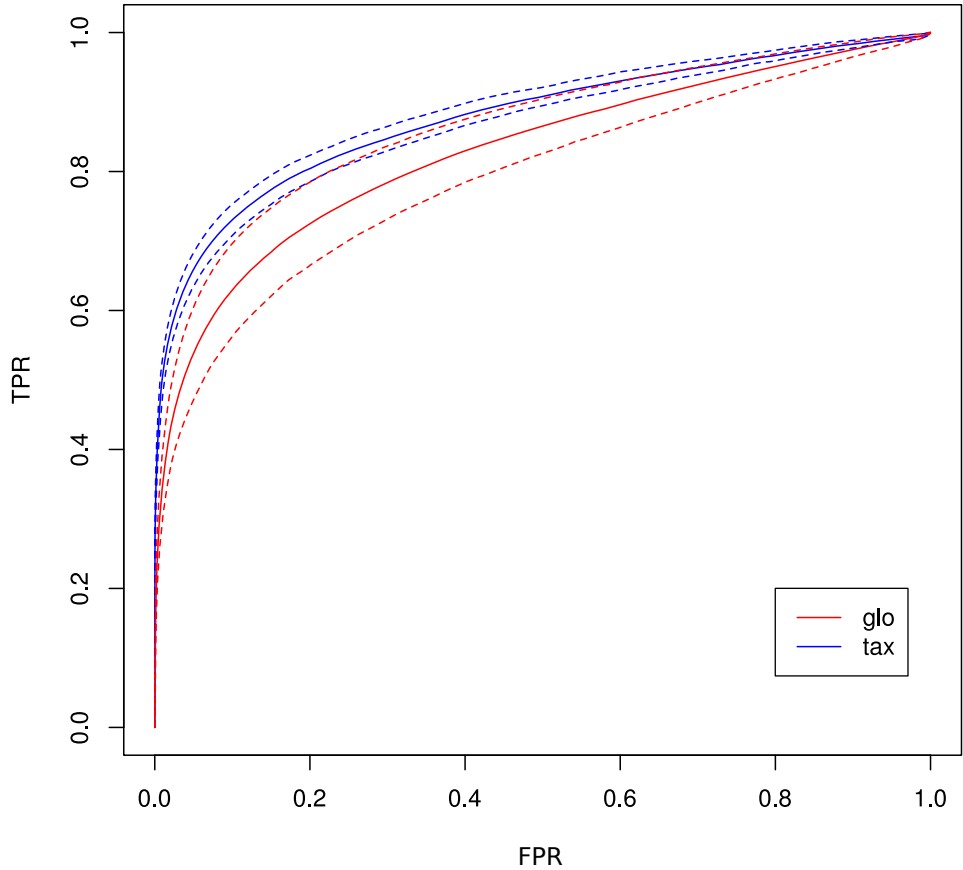

**Figure 4  ROC curves for Simulation II.** Average curve for taxon-specific scaling (blue) vs. average curve for global scaling (red) with false positive rate (FPR) on *x*-axis and true positive rate (TPR) on *y*-axis. Dotted lines above and below indicate the standard deviation for each method. The average area under curve (AUC) is 0.8776 for taxon-specific scaling and 0.8282 for global scaling.

global scaling also shows a higher degree of variation across different simulation runs (Fig. 4 dotted lines).

## Simulation III

With the inclusion of a condition dependent variation of the LS this simulation experiment can be viewed as a worst case study. For global scaling, the observed number of true positives is higher for all data sets compared to Simulation II (see Fig. 5). However, the number of false positive predictions explodes and even exceeds the total number of DEF (500) resulting in average TP and FP numbers of 228 ($\pm$11) and 1,523 ($\pm$78), i.e., $\sim$35% of all features are predicted to be DEF.

In particular, the biggest portion of FP accumulates in features from Org1 and Org2 (see Fig. 5). Inspecting the log2 fold changes (see Fig. 6) a shift from the correct center of 0 upwards and downwards can be observed for Org1 and Org2, respectively. As a result, many DEF are identified with the wrong (opposite) direction and most of the false

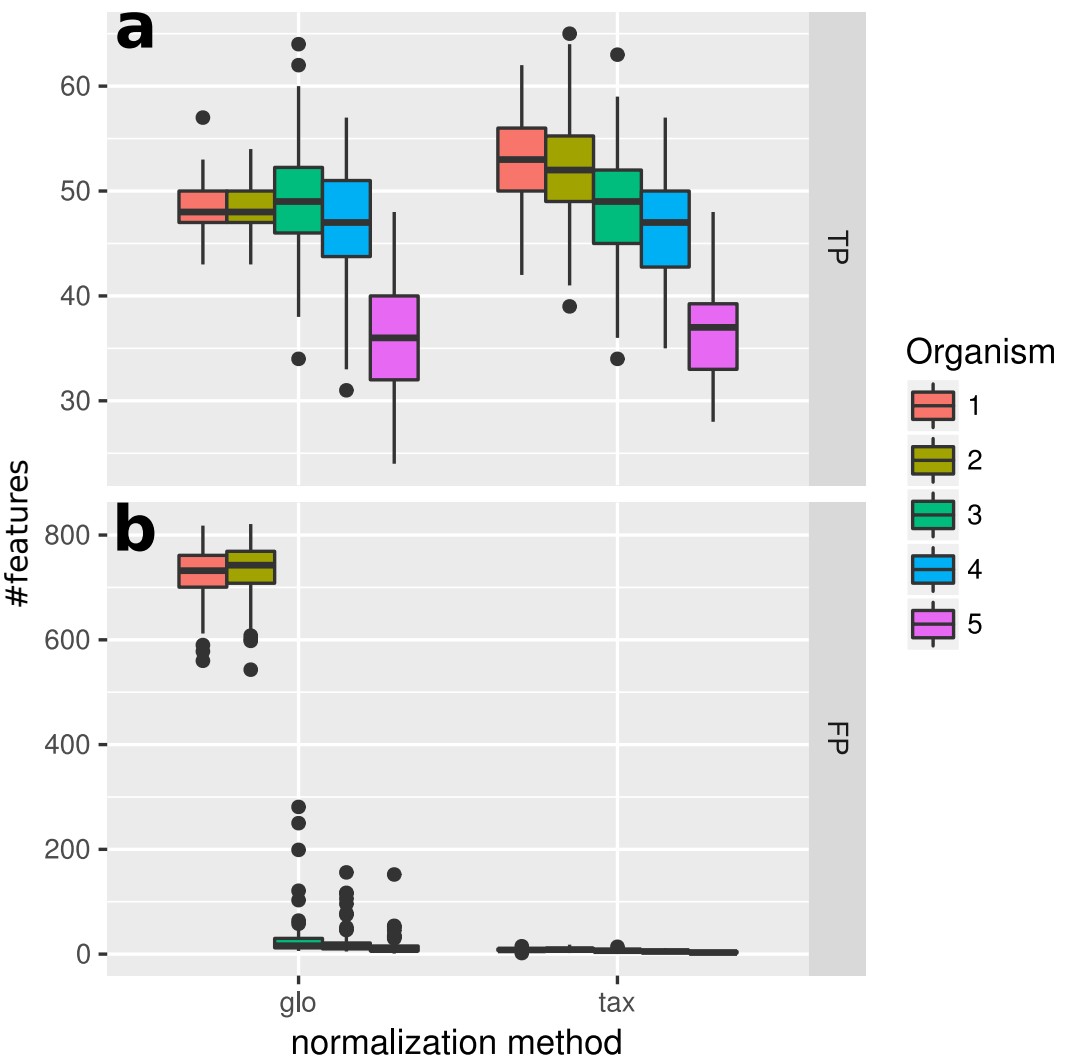

**Figure 5  Simulation III.** Number of true positive (TP) and false positive (FP) features identified with DESeq2 for global ("glo") and taxon-specific ("tax") scaling. Boxplots represent variation over 100 runs of the simulation.

positive detections just reflect the direction of this shift. The results with edgeR and TC normalization show a similarly high FP rate (see File S4: Fig. 5).

As a consequence, the ROC curve collapses for global scaling (see Fig. 7) with an AUC of 0.6396. In contrast, taxon-specific scaling does not suffer from condition-dependent LS variation and the results compare well with those of Simulation I & II showing a similar shape of the ROC curve (AUC: 0.8785). With taxon-specific scaling, the average TP across all species is 237 which corresponds to a sensitivity of ∼47%. For global scaling the total number of predicted DEF (TP + FP) is dependent on the amplitude of the condition dependent LS shift and increases for bigger shifts.

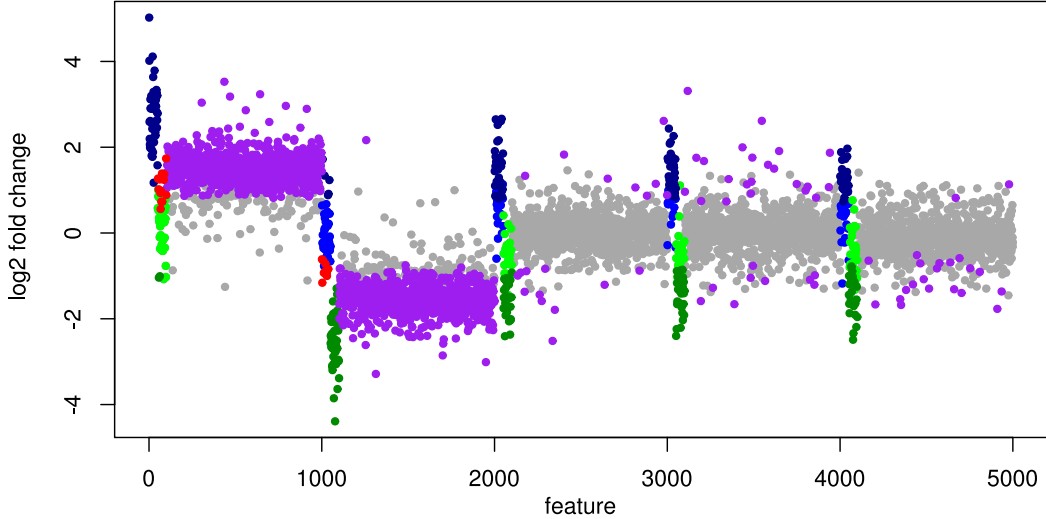

**Figure 6** **Simulation III: Log2 fold changes.** Log2 fold changes of features for global normalization on one example data set. Along $x$-axis, features (dots) are ordered according to five stacked organism profiles, each with 1,000 features of which the first 50 features are "upregulated", and the next 50 features are "downregulated". Gray dots correspond to correctly detected NDE features, light green dots to downregulated features which are missed and dark green dots to correctly identified downregulated DEF. Light blue dots correspond to missed upregulated DEF and dark blue dots to correctly identified upregulated DEF. Red dots mark DEF where global scaling leads to an incorrect direction. Purple dots correspond to NDE features which are incorrectly identified as significant features.

Note that the FP predictions under global scaling arise from a transcriptomics perspective which assumes that all information is represented in the functional variation of organism-specific feature profiles. We admit that for this simulation the term "false positives" can be misleading, because we introduced a systematic condition-dependent LS variation which might as well correspond to a meaningful reaction to a change of experimental conditions. However, the main point here is that with global scaling it would not be possible to distinguish between both kinds of predictions in real data. As shown above, also the number of identified transcriptomic differences (TP) is high and in a typical application of global scaling on mixed counts from different organisms the two types of transcriptional differences will not be recognized.

## Simulation IV

The last simulation only includes two organisms for a better illustration of the effects that arise from a superposition of counts from different organisms. The normalized superimposed counts show a characteristic picture (Figs. 8A and 8B) when looking at the mean counts for condition A ($x$-axis) and B ($y$-axis). While the feature distribution for taxon-specific scaling shows an elliptical distribution with a higher concentration near the diagonal, for global scaling we observe a clustering of features, with a lower density near the diagonal. The two clusters arise from the two organisms, which show a different ratio of condition-specific means, due to different LS under the two conditions. Here we see that it can be problematic to infer the scaling factors which are typically estimated from
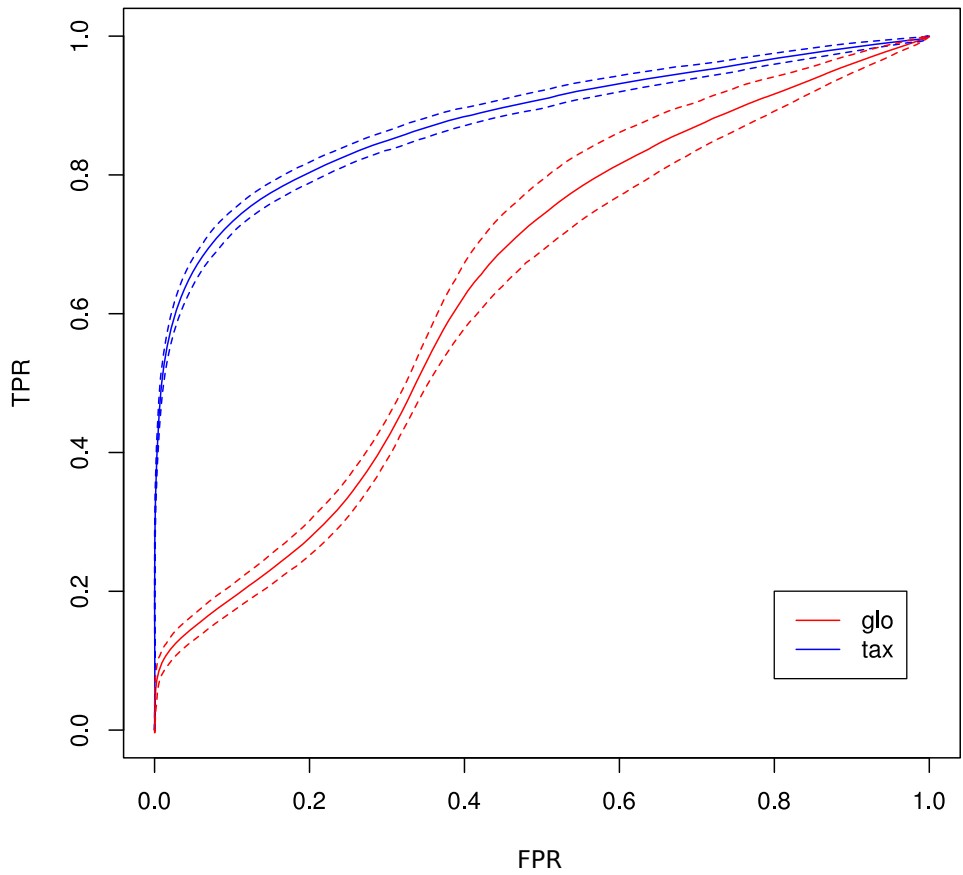

**Figure 7 ROC curves for Simulation III.** Average curve for taxon-specific scaling (blue) vs. average curve for global scaling (red) with false positive rate (FPR) on *x*-axis and true positive rate (TPR) on *y*-axis. Dotted lines above and below indicate the standard deviation for each method. The average area under curve (AUC) is 0.8785 for taxon-specific scaling and 0.6369 for global scaling.

putative NDE features on and near the diagonal. In this case, for transcriptomics tools like DEseq2 and edgeR the assumption of a majority of NDE features is obviously violated.

To illustrate how the superimposed (mixed) counts are obtained, we used arrows to show the contribution of both organisms to the mixed counts for the first 50 features, i.e., all up-regulated DEF from the generated transcriptome profiles (see Figs. 8C and 8D). Each arrow starts at the location of the transcriptomic (mean) counts and ends in the mixed (mean) count. Because we used two organisms, each metatranscriptomic count has two transcriptomic sources. For taxon-specific normalization, most arrows are (nearly) parallel to the diagonal, which indicates that the type of the difference is maintained. For global scaling the arrows form a larger angle and often cross the diagonal. A crossing means that the direction of the difference changes for the mixed feature. A feature that was up-regulated for one transcriptome becomes down-regulated for the metatranscriptome. With global scaling, on average 43.6 crossings occur for the 100 features labeled as DE by the generation process. Only six crossings were observed for taxon-specific scaling (see Figs. 8C and 8D). Thus, the meaning of up and down regulation may completely change with global

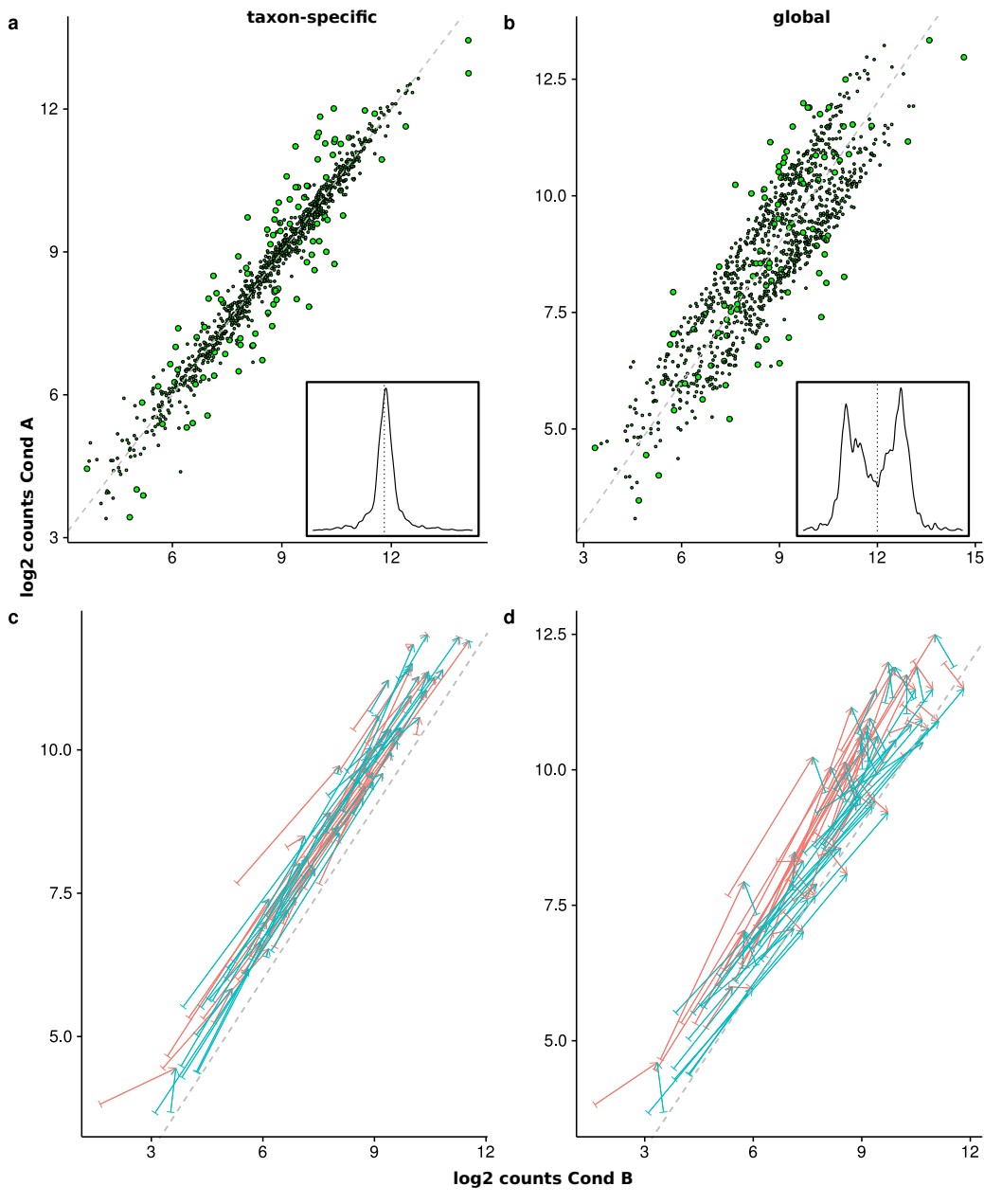

**Figure 8  Comparison of the mean scaled counts between Condition A (Cond A) and Condition B (Cond B).** The figures (A) and (C) show the results of taxon-specific scaling and (B) and (D) the results for global scaling. On the $X$-axis is the log average count per feature in Condition A and on the $Y$-axis the log average count per feature in Condition B. The dashed line represents the balance between counts from Condition A and Condition B (i.e., ideal NDE features). Points show superimposed count data for taxon-specific scaling (A) and global scaling (B). The embedded figure displays the density of the distance of each feature to the dashed diagonal. In (C) and (D) the start of each arrow corresponds to a stacked feature count that is labeled "upregulated" by the data generation process and the color indicates the source organism. The head of each arrow indicates the superimposed count for one of these 50 DEF for taxon-specific (C) and global (D) scaling.

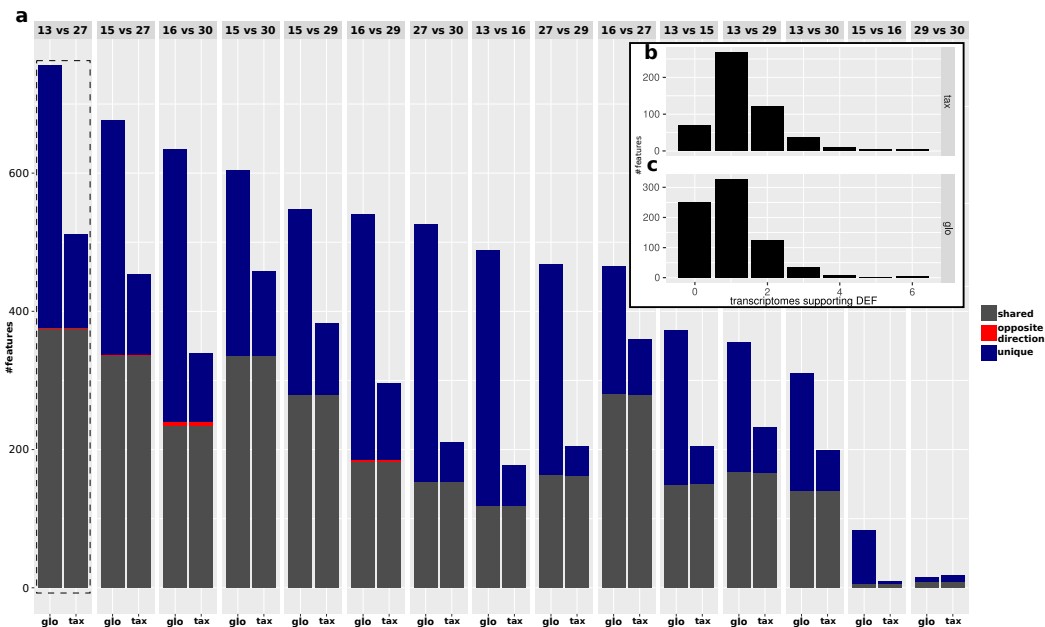

**Figure 9** Predicted DEF for real data (A). Number of significant features from taxon-specific scaling ("tax") and global scaling ("glo") for different condition comparisons. Colors indicate shared significant features with same direction of difference (grey), shared significant features with opposite direction (red) and mutually exclusive features (purple) that are only found to be significant for one scaling method. Histograms for predicted DEF according to the number of single organism analyses that show a significant difference (*x*-axis) with results for taxon-specific scaling (B) and global scaling (C). For example, a high bar at "0" means that many features are found to be significant for the metatranscriptome which are not significant for any of the single transcriptome analyses.

scaling and in large parts the results may contradict the corresponding single transcriptome analyses. This may even rise the question whether the term "differential expression" is still adequate for this kind of difference which arises from a shift of taxonomic abundances.

## Real metatranscriptome data

While the objective of the simulation studies was to evaluate and compare the two normalization approaches in terms of combined transcriptomic DEF, we do not have DE and NDE labels for the analysis of the real data. Therefore we focused on an analysis of the (dis-)agreement in results between both approaches without assessing the detection performance. The analyzed data comprises Pfam counts from 11 organisms, six conditions according to different time points and four replicates per condition. An overview on predicted DEF in all pairwise condition comparisons is shown in Fig. 9.

For both approaches, the number of predicted DEF peaked at "day 13" vs. "day 27" with 512 and 756 significant features for taxon-specific and global scaling. The number of DEF was low when conditions close together on the time line were compared ("day 15" vs. "day 16" or "day 29" vs. "day 30").

For global scaling, the number of predicted DEF was generally higher than for taxon-specific scaling. For some of the comparisons, the number of extra predictions under

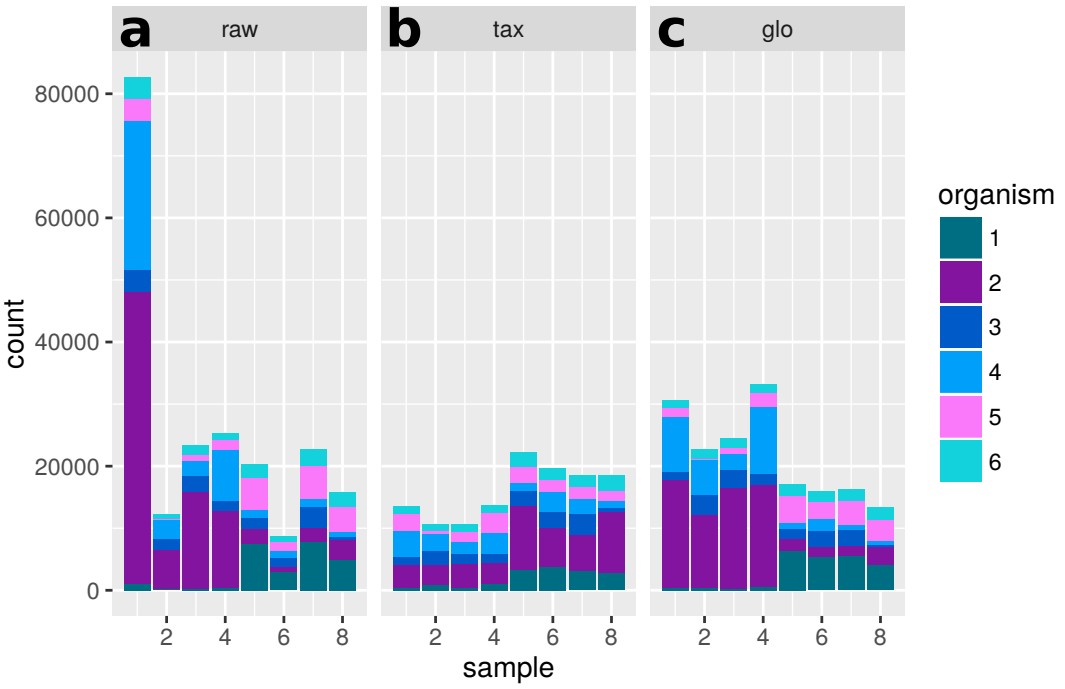

**Figure 10** **Single feature analysis (PF12667).** Stacked bars in three parts ($x$-axis) show organism-specific counts before scaling ("raw"), after taxon-specific scaling ("tax") and after global scaling ("glo"). Significant feature with opposite direction for the two scaling methods. Taxon-specific and global scaling result in adjusted $p$-values $6.91e^{-5}$ and $1.33e^{-5}$, respectively. The log2 fold change for condition A in comparison to condition B is 0.70 for taxon-specific scaling and $-0.82$ for global scaling.

global scaling was even higher than the number of shared predictions (see Fig. 9). The high number of extra predictions observed with global scaling is especially prevalent for the comparisons "day 15" vs "day 16" with 16 times more extra predictions than predictions shared with taxon-specific scaling and "day 13" vs "day 16" with three times more extra predictions than shared predictions.

We also compared the results of the mixture analysis for global scaling and taxon-specific scaling to the transcriptome analyses of the individual organisms. For 10 of the 15 comparisons, the majority of predicted DEF with global scaling were not predicted as DEF in any transcriptome (see File S4: Fig. 6). In contrast, with taxon-specific scaling only the comparisons "day 29" vs "day 30" and "day 15" vs "day 16" show a higher fraction of significant features not predicted as DEF in the transcriptomes. These two comparisons are also the ones with the smallest total number of predicted DEF. When taking into account the direction of the differential expression, the number of DEF predicted by both methods but with contrary regulation direction is low. Here, comparison "day 16" vs "day 30" shows a maximum of 5 DEF with an opposite direction. An example of a feature that shows a contrary result under the two scaling methods is shown in Fig. 10.

### Analysis of "day 13" vs "day 27"

As described in the original study, "day 13" and "day 27" each correspond to the final day of a particular diet. Because the number of predicted DEF for both scaling methods (global and taxon-specific) was the highest here, we analyze the results for this comparison in more detail.

We mapped the Pfam-annotated features which were predicted as DEF for global scaling and taxon-specitic scaling to Gene Ontology (GO) terms and compared the results. With taxon-specific scaling 250 GO terms with at least one DEF mapping were identified, while global scaling resulted in 311 GO terms. GO terms associated to biological processes with a high agreement between the two methods were for example cellular amino acid metabolic process where both methods identified seven of the nine associated Pfams as DEF and carbohydrate metabolic process with 11 DEF shared between taxon-specific scaling and global scaling. In this category, taxon-specific scaling and global scaling predicted five additional DEF uniquely.

GO terms with predicted DEF from taxon-specific scaling alone included magnesium ion binding and fucose metabolic process with three predicted DEF each. For global scaling alone, we found that DNA modification, molybdopterin cofactor biosynthetic process, and RNA modification with 3, 2 and 2 predicted DEF respectively (see File S6 for a complete list).

*Extra predictions.* In the condition comparison "day 13" vs "day 27", both normalization approaches shared 376 DEF. With taxon-specific scaling, 136 extra predictions were observable while global scaling resulted in 380 extra predictions. We then compared the results of both scaling methods to the single organism transcriptome analyses where a total of 1,302 features had been found to be DEF at least for one transcriptome. Global scaling and taxon-specific scaling resulted in 252 and 69 predictions of DEF that were insignificant in all single analyses (see Fig. 9). Within these predictions both methods shared an overlap of 53 features.

The fraction of shared DEF predictions between the two scaling methods is lower if the DEF are supported by a smaller number of transcriptome analyses. For features supported by one transcriptome the agreement was ~48%, increasing to ~56% for two transcriptomes. In the range of three to six supporting transcriptomes, the agreement increases to ~59%, ~80%, 100% and 100% respectively. While the relative agreement between taxon-specific and global scaling increases, the total number of features supported by multiple transcriptomes decreases (File S4: Fig. 6).

The results indicate that with global scaling we encounter the same problem that we have already outlined in the simulations. As shown above, on real data the number of shared DEF predictions between the two scaling methods can be as high as the number of extra predictions that result from global scaling. As also indicated by the following scaling factor analysis there is strong evidence that these extra predictions result from taxonomic abundance variations. Again we argue, that without taxonomic separation and further analysis, with global scaling it is impossible to distinguish differences that result from taxonomic abundance shifts from differences that arise from organism-specific changes of the transcriptome.

*Scaling divergence.* In the simulations, the differences in the estimated scaling factors for the single organism profiles in comparison to the actual scaling factors were low with a scaling divergence of ∼0.01. In Simulation II we found the scaling deviation to be high in general with global scaling (File S4: Fig. 3) and in Simulation III which showed the highest divergence we observed a drastic increase of predicted DEF with global scaling when two organisms with condition-dependent abundance shifts were combined.

To further investigate the increased number of DEF predicted by global scaling, we compared the scaling factors estimated for the single organism profiles with the estimated scaling factors for the global normalization in the comparison "day 13" vs "day 27". For several organisms, a pattern emerged which shows a condition specific scaling divergence (Fig. 11). While the scaling factors for one condition are relatively small, the scaling factors in the other condition are relatively high. As a result, global scaling leads to a condition-dependent divergence which may explain the increased prediction of DEF because the normalization introduces a shift between the two conditions. The observed scaling divergence is also in agreement with the organism abundance for each sample (see File S4: Fig. 7).

A single scaling factor per sample is especially problematic for features, which mainly comprise counts from one organism or when counts from mixed organisms with the same scaling shift are analyzed. For a quantification we determined the number of features, for which counts from a single organism (or the mixture of organisms with the same scaling divergence direction) exceed 80% of the normalized counts for that feature. For extra predictions obtained only with global scaling without evidence from the transcriptome analyses, the counts from a single organism are the main contribution for 82 of 199 features. Additional 43 features are predicted from the summed counts from organisms with the same scaling shift. An example of a feature that shows a dominating single organism is shown in Fig. 12.

## DISCUSSION

The simulations allowed us to highlight the effects of the two normalization approaches that can lead to results with a varying degree of divergence between taxon-specific and global scaling. Here, we could show that organism-specific transcript abundance and its variation between and within conditions is the key factor for the divergent results.

Other important factors are the accuracy of the functional and taxonomic annotation and the number of observed features with sufficient counts to estimate the scaling factors. In this regard, the simulations correspond to idealized conditions which cannot reflect the difficulties of a real data analysis. The number of organisms was chosen with respect to a clear visualization of results. In principle, an increase of the organism number does not add any effects that could not be observed with a lower complexity.

All simulations were designed to demonstrate the very different nature of the two normalization approaches. Taxon-specific scaling is closely related to the transcriptomic approach to differential expression analysis and can be viewed as a possible generalization of the single-organism setup to a multi-organism analysis. It equalizes all variations of the taxonomic composition of transcripts across samples to identify functional effects

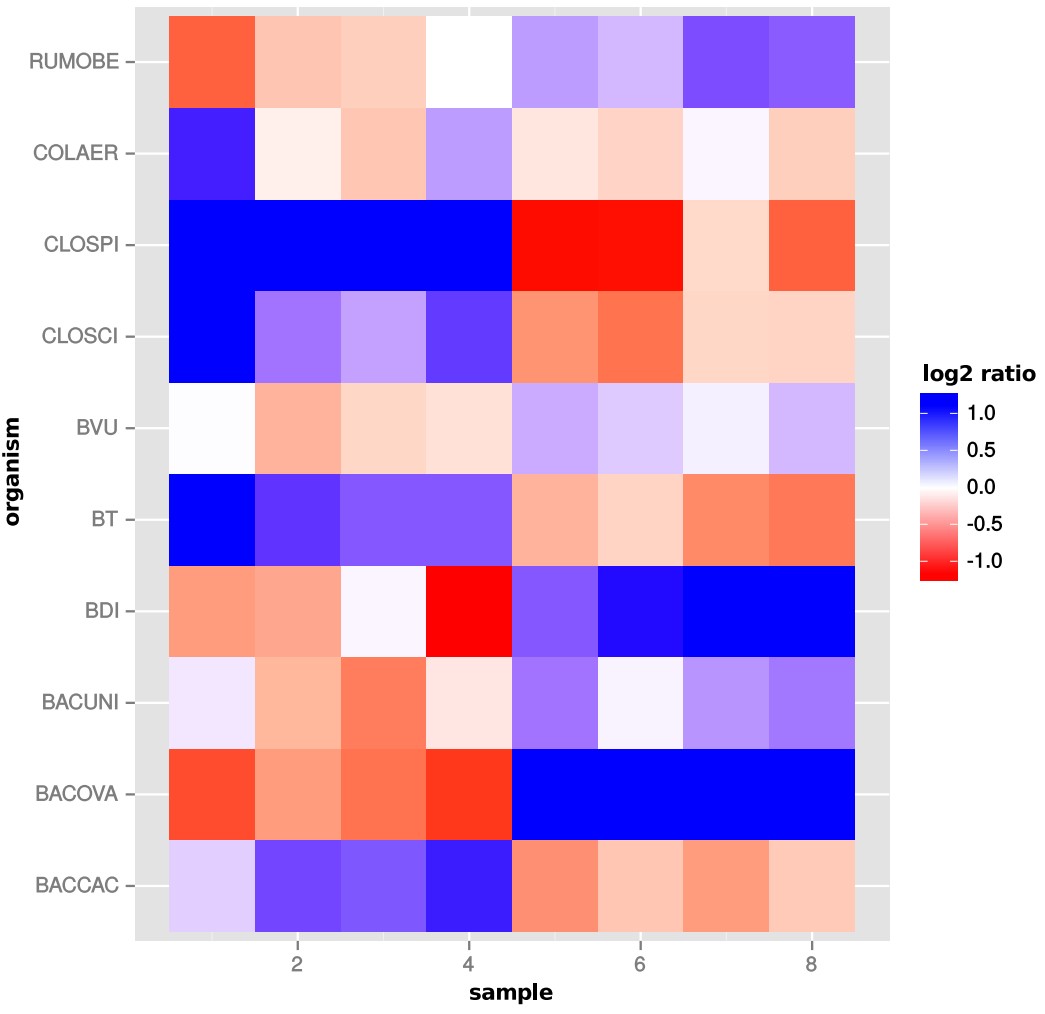

**Figure 11  Global scaling condition bias.** Direction of global scaling divergence in terms of the log2-ratio of scaling factors from transcriptomic and global scaling. Results for different organisms in the comparison of ''day 13'' vs. ''day 27''. For symmetry of the color range the negative log2-ratio was capped at −1.25, with divergence values below that threshold showing the same color (blue). Samples 1–4 are from condition A and samples 5–8 from condition B. For the species name abbreviations see File S2: Table 1. *D. longicatena DSM 13814* was not observed in that particular condition comparison.

that are consistent across different organisms. If the functional profiles do not show systematic variation between conditions, no meaningful differences will be found with taxon-specific scaling.

In contrast, global scaling assumes that variations of the taxonomic composition are meaningful and important for differential analysis. Depending on the data, this assumption may be correct and then many of the false positives for global scaling in Simulation III may actually correspond to true positives. The objective of the simulations was to compare the two normalization approaches under the assumption that the relevant information is in the variation of the organism-specific functional profiles, similar to the transcriptomic approach. In contrast, the taxonomic variations were not assumed to be indicative of the

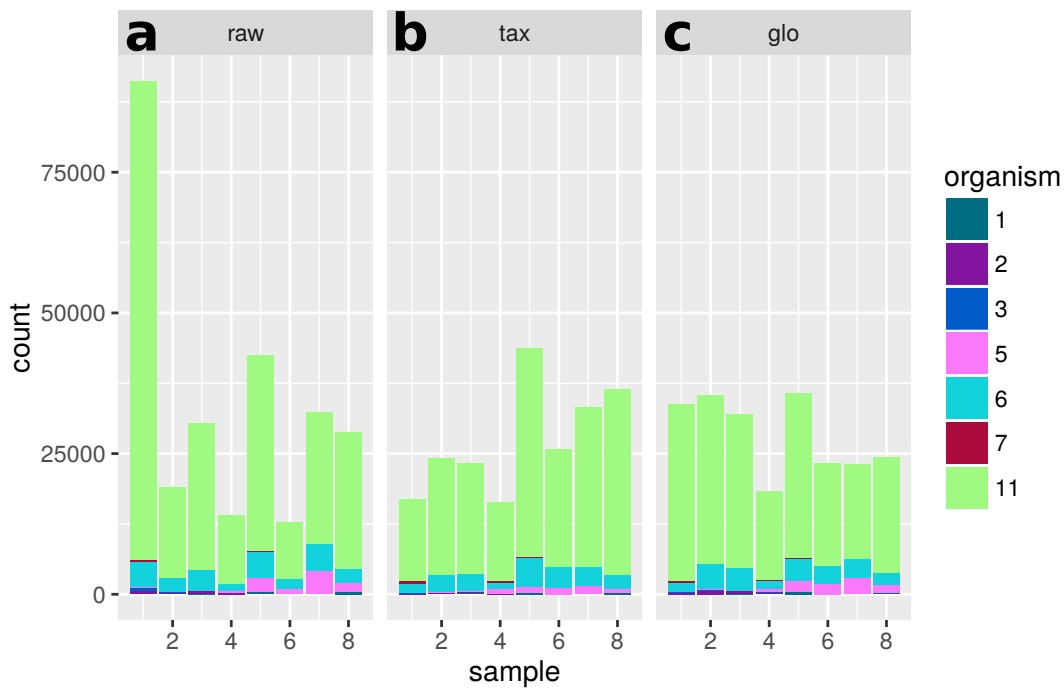

**Figure 12** **Single feature analysis (PF07881).** Stacked bars in three parts ($x$-axis) show organism-specific counts before scaling ("raw"), after taxon-specific scaling ("tax") and after global scaling ("glo"). Loss of significance due to (mis)scaling of profiles from mainly one organism. Feature is significant for organisms Org2, Org5 and Org11 in transcriptome analysis. Taxon-specific and global scaling result in adjusted $p$-values $1.79e^{-3}$ and $0.66$, respectively.

experimental conditions and might for instance result from sample preparation. We chose this setup because it reveals a potential complication of the global scaling approach. As the main problem, the effects of taxonomic and functional variations will always be mixed up in the results when applying global scaling. Without further analysis, it is impossible to determine whether a predicted DEF results from a taxonomic or from a functional variation. Another problem with global scaling has been shown in Simulation IV, where the underlying model assumptions for the normalization tend to be violated. With an increasing number of DEF it becomes increasingly unclear which features should be used as a reference for estimation of the scaling factors. Therefore, if taxonomic variation is informative it might be a better strategy to estimate the species composition for each sample and then compare the taxonomic profiles for the identification of significant differences. In fact a differential composition analysis as used in metagenomics (*McMurdie & Holmes, 2014*; *Weiss et al., 2017*) will well complement a differential expression analysis based on taxon-specific scaling.

It is important to realize that while taxon-specific scaling negates library size variation of the organisms between samples, it retains the overall taxonomic composition of the observed transcripts. Therefore, the feature counts of rare organisms will have less impact on the combined metatranscriptome data than feature counts from highly abundant organisms. The minimum required coverage of an organism for reliable estimation of

the scaling factors highly depends on the data. For rare species it is possible that only a few highly expressed features are observed. As a special case, organisms only observed in one condition cannot be included in the differential expression analysis of the combined counts for taxon-specific scaling. Therefore, the user has to check what fraction of the features can actually be used in the analysis and if identified DEF could be the result of the exclusion of features from organisms not observed in both conditions. With global scaling such single-condition organisms can be included but without taxonomic separation it can become difficult to explain a possibly large number of differences from a change of experimental conditions.

For taxon-specific scaling our approach relies on techniques implemented in tools that have been developed for differential expression analysis in transcriptomics. As a consequence, our method also inherits the assumptions and requirements of these tools for the normalization of transcriptomic count data. In general, both DESeq2 and edgeR require a sufficient number of NDE features to reliably perform the normalization. In particular, both methods expect the number of DEF to be lower than the number of NDE features. Also the proportion of up- and down-regulated features may have an influence on the detection of DEF. When the ratio is shifted towards one regulation direction, the detection performance may decrease. Other normalization methods might provide better results under these conditions (*Soneson & Delorenzi, 2013*). The applicability of taxon-specific scaling inherently depends on the accuracy of the methods that are used for taxonomic binning and feature annotation. Among the factors that impact the accuracy are the complexity of the analyzed metatranscriptome and the evolutionary distance between the observed organisms and organisms in reference databases. For highly complex communities and for communities with only distantly related organisms in the reference databases, additional sequencing of the metagenome will be highly beneficial to improve the results.

The optimal situation for taxon-specific scaling would enable the separation of all feature profiles from different organisms. This organism-specific separation can be viewed as the most conservative approach, as in principle it would be sufficient to separate all differing expression profiles. Therefore, if organisms show the same transcriptional activities, the normalization of their combined count data is possible without problems. For example, the metatranscriptome might include multiple strains of the same species occurring simultaneously but showing the same activities.

The choice of features is also an important factor to consider before the experiments are performed. The transformation of the organism-specific genes to a common feature space, such as the Pfam protein domain categories, allows us to first normalize and then sum up the features of different organisms to obtain a superimposed count matrix. The mapping can also include the assignment of multiple genes from the same organism to a single feature representation. As a consequence, it is no longer possible to identify differential expression with respect to a single gene. This limitation is inherent to many metatranscriptome studies and does not depend on the chosen normalization approach. Again, parallel sequencing of the metagenome might be considered, to obtain a higher resolution. Then, a combined

assembly and binning strategy can yield the required draft genomes as reference to realize the highest taxonomic and functional resolution for differential expression analysis.

## CONCLUSIONS

Differential expression analysis in metatranscriptomics is challenging. Metatranscriptomic count data from RNA-Seq experiments show two main modes of biological variation. The functional composition of transcripts reflects the activity of organisms and systematic changes might indicate a metabolic response to experimental conditions. The taxonomic composition of transcripts can change as well and a change may not necessarily be explainable in terms of controlled experimental conditions. In contrast to metagenomics, in metatranscriptomics the questions "who is there?" and "what are they doing" are not necessarily connected and should be answered separately for a clear interpretation of results. If the two questions are not separated, just from a significant difference it is not possible to decide whether this difference goes back to a variation of taxonomic or functional profiles or if it reflects both kinds of variation. Our approach to normalization of metatranscriptomic data eliminates the influence of taxonomic variations from functional analysis. For realization of this approach, the metatranscriptome needs to be decomposed to normalize the organism profiles independently. Then the metatranscriptomic count data can be recombined from the normalized profiles to look for any general tendencies in the superimposed count data. If differential expression tools are directly applied to the metatranscriptomic count matrix, the interpretation of results can be difficult. Our simulations indicate that it is easily possible to obtain a large number of putative functional differences which just arise from taxonomic abundance variations across samples. We would like to point out that our findings do not affect metatranscriptome studies that just aim to analyze the functional repertoire from RNA-Seq data. The question which functions or genes are expressed is much easier to answer than the question what is the functional response to a change of experimental conditions. However, it is important to note that our results do not only apply to the classic two conditions setup that we used throughout our study. Also for multiple conditions and time series our normalization approach will be beneficial to separate functional from taxonomic trends in metatranscriptomic count data.

### Funding

This work was funded by the University of Goettingen and DFG (Project: "Computational models for metatranscriptome analysis", Me3138). There was no additional external funding received for this study.

### Grant Disclosures

The following grant information was disclosed by the authors:
University of Goettingen.
DFG.

## Competing Interests

The authors declare there are no competing interests.

## Author Contributions

- Heiner Klingenberg conceived and designed the experiments, performed the experiments, analyzed the data, wrote the paper, prepared figures and/or tables, reviewed drafts of the paper.
- Peter Meinicke conceived and designed the experiments, analyzed the data, wrote the paper, reviewed drafts of the paper, statistical modeling.

## Data Availability

The R code for count data normalization and the count data used for evaluation have been provided as Supplemental Files.

## Supplemental Information

Supplemental information for this article can be found online at http://dx.doi.org/10.7717/peerj.3859#supplemental-information.

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
