# Peer review of "How to normalize metatranscriptomic count data for differential expression analysis"

_PeerJ, doi:10.7717/peerj.3859_

## Round 0.1 · original submission · Major Revisions

Please address the comments from both reviewers, particularly reviewer 2. Be more clear about the context, assumptions and limitations of the proposed methods. The method is useful when the bacteria mixture is thought to be a confounding factor, and we have ways to separate the counts by bacteria.

One additional minor comment:

"The median is highly robust, with a breakdown point of 50% and
therefore the estimator can be used if at least half of the data corresponds to NDE features." This statement is not accurate. Median is robust when the majority of the features do not change.

Reviewer 1 ·

Basic reporting

Some text descriptions are really confusing. The mathematical formulas are not clear. Overall, the English writing should be improved.

Experimental design

The simulation steps are not described clearly.

Validity of the findings

Though the simulation results suggest that the proposed taxon-specific scaling outperforms global scaling, the real data results are not enough to support the claim.

Additional comments

The authors claim that calculating scaling factors on the level of taxon instead of on the level of global, the performance of differential analysis will be improved. I have some comments on the follows,

1. Some formulas are confusing. For example, formula (3). The estimated scaling factors are only determined by feature i, which is obviously not correct.

2. Some statements are confusing. For example, what does “predictions“ means in line 40? Do the predictions mean DEFs? In line 124, which is “organism-specific abundance”? What is the difference between organism-specific and taxon-specific?

3. The simulation steps are not described clearly. How did the abundance generated, based on what kind of generative model? How did the count matrix of different organism combined together? What is the proportion of each organism?

4. In the real data comparison on page9 13-15, the gold standard for DEF for both global scaling and taxon-specific scaling. The comparisons of the numbers of DEFs identified of both approaches cannot support the claim one outperforms the other, and vice versa. More identified DEFs could be false positive, and few identified DEFs could be lacking of statistical power. The authors should use other criteria to justify taxon-specific scaling is better than global scaling.

Reviewer 2 ·

Basic reporting

In this manuscript, the authors propose a strategy for normalizing metatranscriptomics data for sequencing depth in the context of differential analysis. They propose the ‘taxon-specific scaling’ to take into account the structure of the metacommunity. This strategy consists in separating the dataset into one dataset per organism, to normalize them independently with standard method such as the method proposed by Anders et Huber in the DESeq2 Bioconductor package and to analyze a combined dataset. They compare this strategy with the ‘global scaling’ one which considers the whole dataset forgetting the notion of species. They study the interest of their strategy, both in simulations and in a real dataset.

The introduction will benefit from a more detailed description of the context of normalization for differential analysis. Are standard strategies for transcriptomic directly applicable for metatranscriptomics ? but also are those for differential abundances from metagenomics data applicable for metatranscriptomics (see Weiss et al. 2017).

The manuscript is not so easy to read for different reasons :
1/ some essential terms essential such as ‘normalization’ or ‘library size’ are not well-defined and need to be explained in the metatranscriptomics context.
2/ it subsists some redundancy (normalization part of the introduction and taxon-specific scaling and global scaling’ paragraph of the materials and methods section).

In order to clarify the manuscript, I suggest to present normalization as between-sample normalization and to define clearly what is the library size. In general, for normalization purpose, the library size is for each sample the total number of sequences which will be analysed. One key point not mentionned here is that normalization depends on the biological question. The main two biases we would like to correct in metatranscriptomics is the composition of metacommunity and the sequencing effort. Both may vary with sample. Depending on the biological question only a correction for sequencing depth or for both is needed. It would be clearer to separate the two and not to speak about ‘organism-specific library size’. I suggest for Simulation II ‘modification in the abundance of species present in the metacommunity’ instead of ‘with library size variation’. The interpretation of the results of differential analysis will depend on the question and on the normalization step and a feature statistically significant can be result of difference in expression or/and in the abundance of the species it comes from.


I suggest, for my second remark, to merge the normalization paragraph of the introduction with the paragraph in materials and methods. And in this section to distinguish in different paragraphs i) the taxon-specific scaling, ii)the global scaling, and only in a third part to discuss about the way to compute scaling factor (using DESeq2 for example).

Experimental design

The normalization of metatranscriptomics data is a great challenge and the question of when and how to take into account the variations in species abundance is an important question.
The authors explore several interesting scenarios. Nevertheless a real metatranscriptomics study contains generally more than 5 or 12 organisms and most of them are uncultivable. Would the authors discuss this point and the application to metagenomic species ?

I thank the authors for providing the real raw data and a script with R code for simulation and DESeq2 analysis. Nevertheless, I would appreciate to have a minimal script that downloads the data and either simulated data nor script with simulated parameters in order to reproduce the results of all the analyses.


When using the test.main function or the DESeq2.tax.specific function, it results on a matrix of results with 1000*6 samples and not 5000 features as noted l.316. Could the authors explain why? Is it because they sum over all the organisms?

L156 the authors wrote ‘There are several reasons why the analysis of the recombined metatranscriptome data can be useful ; first of all, the statistical power of organism-specific tests may be low due to decreased counts.’ I do not agree : if the analysis is done organism by organism, the correction for multiple testing will be less stringent (1000 tests instead of 5000).

Validity of the findings

No comment

Additional comments

I regret the absence of discussion on
1/ assumptions for normalization which are i) the great majority of the genes are DE and ii) balance between up and down regulated features. Could the author develop this point and the cases other way to compute scaling factors (ie instead of DESeq2 normalization) is necessary ?
2/ validity of the proposed strategy. Could the authors discuss
i) on the situations where it is really possible to obtain counts per organisms, and which minimal coverage is needed for each organism, etc. ?
ii) when it is reasonable to sum over features?
iii) the case of a great difference in metacommunity diversity, e.g. several organisms absent in one condition?


In the conclusion, the authors write l. 502-506 « Normalization of metatranscriptomic data must have the goal to eliminate the influence of taxonomic variations from functional analysis. (…) We argue that for a correct normalization the metatranscriptome needs
to be decomposed to normalize the organism profiles independently. Then the metatranscriptomic count
data may be recombined from the normalized profiles to look for any global trends in the superimposed count data.’ How to superimpose count data is not well explained . Is it always straightforward ?



Comments for the authors

The reference for DESeq2 is (Love et al. 2014) and in l.186 the authors need to cite Bionconductor and the version they used. One way to obtain references for R packages is to tape « citation(« pkgname ») in the R console.

The reference for R is the following one :
R Core Team (2017). R: A language and environment for statistical computing. R
Foundation for Statistical Computing, Vienna, Austria. URL
https://www.R-project.org/.


Minor revisions

L50 'transcriptome' is misspelled as 'transciptome'.
L133 ‘separated’ is misspelled as ‘seperated’.
L211 ‘library’ is mispelled as ‘libray’.

l.175 the right name for the normalization method implemented in edgeR is TMM (Trimmed-Mean of M-Values) (Robinson and Oschlack 2010) (l175 , l317,318, 347)

The abbreviation used for differentially expressed features is DEF (l.196) but either DE nor DE is used all along the manuscript. Please choose one and be homogeneous.

Figure 9. the size of the legend and labels is too small. One way to gain place may be to replace « 13d vs 27d » by « 13 vs 27 »

Reference
Weiss,S., Xu,Z.Z., Peddada,S., Amir,A., Bittinger,K., Gonzalez,A., Lozupone,C., Zaneveld,J.R., Va ́zquez-Baeza,Y., Birmingham,A.
et al. (2017) Normalization and microbial differential abundance strategies depend upon data characteristics. Microbiome, 5, 27.

---

## Round 0.2 · Minor Revisions

Please address the minor comments of the reviewer. After that, the manuscript will be suitable for publication.

Reviewer 2 ·

Basic reporting

no comment

Experimental design

no comment

Validity of the findings

no comment

Additional comments

In this manuscript, the authors propose a strategy for normalizing metatranscriptomics data for sequencing depth in the context of differential analysis. They propose the ‘taxon-specific scaling’ to take into account the structure of the metacommunity. This strategy consists in separating the dataset into one dataset per organism, to normalize them independently with standard method such as the method proposed by Anders et Huber in the DESeq2 Bioconductor package and to analyze a combined dataset. They compare this strategy with the ‘global scaling’ one which considers the whole dataset forgetting the notion of species. They study the interest of their strategy, both in simulations and in a real dataset.

I thank the authors for the revised version of their manuscript, which considers all my previous comments and suggestions.

Just few typos and one minor comment:

peerj-17882-Additional_File_4:
Fig 4 legend: ‘library’ is mispelled as ‘libray’.

Fig 6 legend : the color box with « white », « black » and « blue » is inexact and not useful.

I tested the code in the peerj-17882-Additional_File_3.R file on the real dataset and I have got the following error message
« Error in `rownames<-`(`*tmp*`, value = colnames(countData)) :
duplicate rownames not allowed »
I do not use the same versions of DESeq2 and R (R 3.4.0 and DESeq2_1.16.1). It may be the origin of this error. This error appears using the DESeq2.norm.mat function line 384
YMat <- DESeq2.norm.mat(XMat, cond.vec,type.vec)
A way to avoid it is to check that colnames of XMat are not duplicated before using the function and rename the columns if necessary.

---

## Round 0.3 · accepted · Accept

All the minor comments have been addressed. The manuscript is ready for publication.